# Assessment of Thoracic Pain Using Machine Learning: A Case Study from Baja California, Mexico

**DOI:** 10.3390/ijerph18042155

**Published:** 2021-02-23

**Authors:** Veronica Rojas-Mendizabal, Cristián Castillo-Olea, Alexandra Gómez-Siono, Clemente Zuñiga

**Affiliations:** 1School of Engineering, CETYS Universidad, Mexicali 21259, Mexico; alexandra.siono@cetys.edu.mx; 2School of Medicine and Psychology, Autonomous University of Baja California, Tijuana 22800, Mexico; 3General Hospital of Tijuana, Tijuana 22000, Mexico; zclemente@hotmail.com

**Keywords:** machine learning, thoracic pain, tree classification, cross-validation

## Abstract

Thoracic pain is a shared symptom among gastrointestinal diseases, muscle pain, emotional disorders, and the most deadly: Cardiovascular diseases. Due to the limited space in the emergency department, it is important to identify when thoracic pain is of cardiac origin, since being a symptom of CVD (Cardiovascular Disease), the attention to the patient must be immediate to prevent irreversible injuries or even death. Artificial intelligence contributes to the early detection of pathologies, such as chest pain. In this study, the machine learning techniques were used, performing an analysis of 27 variables provided by a database with information from 258 geriatric patients with 60 years old average age from Medical Norte Hospital in Tijuana, Baja California, Mexico. The objective of this analysis is to determine which variables are correlated with thoracic pain of cardiac origin and use the results as secondary parameters to evaluate the thoracic pain in the emergency rooms, and determine if its origin comes from a CVD or not. For this, two machine learning techniques were used: Tree classification and cross-validation. As a result, the Logistic Regression model, using the characteristics proposed as second factors to consider as variables, obtained an average accuracy (μ) of 96.4% with a standard deviation (σ) of 2.4924, while for F1 a mean (μ) of 91.2% and a standard deviation (σ) of 6.5640. This analysis suggests that among the main factors related to cardiac thoracic pain are: Dyslipidemia, diabetes, chronic kidney failure, hypertension, smoking habits, and troponin levels at the time of admission, which is when the pain occurs. Considering dyslipidemia and diabetes as the main variables due to similar results with machine learning techniques and statistical methods, where 61.95% of the patients who suffer an Acute Myocardial Infarction (AMI) have diabetes, and the 71.73% have dyslipidemia.

## 1. Introduction

Thoracic pain is one of the generally most relevant factors in people with cardiovascular problems at risk of heart attacks. However, despite its relevance in this area, chest pain may be an indicator of some other pathology not related to CVD. In 2015, the WHO (World Health Organization) recorded 17.7 million deaths related to CVD, where 42.8% were due to coronary heart disease and 36.15% to cerebrovascular accidents [1]. While the World Heart Federation in 2017 reported that in Mexico, 77% of deaths were due to NCD (Non-Communicable Diseases), where 24% of these were caused by CVD [2]. In 2018, the INEGI (National Institute of Statistical Geography) reported in Baja California 149,368 cases of death from CVD, where ischemic diseases represented 72.7%, while hypertensive diseases were 15.9%; the rest were split between pulmonary vascular disorders and acute rheumatic fever, among others [3].

Since CVDs are involved with a large percentage of the causes of death in Baja California, a decision was made to analyze a database with information from 258 patients provided by Medica Norte with variables such as Edad, Género, Fumador, HTA, Dyslipidemia, Diabetes, ERC (Cr basal), Suma FRCV, C. Isquémica previa, PPT, Rangos PPT, Tipo dolor, TnT Ingreso, TnT Curva (4 h), ECG, Tipo Alteración, TC > 100, IC, Alta precoz, UDT, Ingreso, Ergometría, Eco stress, Cate, Angio TAC, IAM, Revascularización (See Appendix A); and thus, with the help of Orange, data analysis was carried out to find which biochemical markers or habits are mostly related to thoracic pain of cardiac origin, to more accurately locate the risk factors involved in development of a cardiac event and dismiss as an emergency those patients with chest pain who do not meet the conditions established for the development of CVD. With these results, a proposal for second parameters to take into account in emergency rooms is produced to avoid possible deaths caused by thoracic pain.

For this analysis, two variables based on troponin were considered, since it is in charge of establishing the frequency of cardiac muscle contraction, which, when affected by a heart attack, is released and can be used as a bio indicator [4]. According to a 2019 study, Troponin has a positive predictive value of 62%, while its negative predictive value is 93% for cardiac lesions [5]. Therefore, the first variable was TnT Ingreso, where troponin levels were measured in the blood of patients on arrival at the emergency room, and the second was TnT curve (4 h), which are the levels of troponin found in the blood of admitted patients four hours later.

When a patient arrives at the emergency room with chest pain, he is evaluated with an exam known as PreTest Probability (PPT), which helps choose the most accurate method of analysis to determine the type of pain in the patient. This PreTest consider variables like gender, age, and some symptoms such as typical angina, atypical angina, or non-anginal pain. Later, depending on the values of these variables, a percentage is established that can be part of one of the four ranges used, and this range will determine the probability that the pain present is due to CVD or not [6].

Among the conventional predictive methods to assess the etiology of thoracic pain are the SCORE (Systematic Coronary Risk Evaluation), ASCVD (AtheroSclerotic CardioVascular Disease) Risk Estimator, and Framingham. The SCORE method is adapted from the guide for CVD prevention in 2016 carried out by a project with the same name, which is based on the calculation of risk factors for the prediction of possible CVDs at 10 years in European patients [7]. On the other hand, ASCVD Risk Estimator evaluates the risk that the patient has of atherosclerosis since this disease affects the arteries causing CVD. While the Framingham method is the most widely used and oldest, since it dates back to 1948, the risk of CVD using this method is calculated by assigning a value to variables related to the patient’s condition and subsequently making a summation that will indicate the risk of developing CVD within 10 years [8].

Nowadays, machine learning technologies, deep learning, and artificial intelligence have been a meaningful tool for the healthcare industry. Thus, its classification and patterns recognition capabilities for applications enable the image processing for treatable diseases diagnosis. In addition, predictions based in mathematical models algorithms using databases to classify different diseases related with a specific system and variables correlation to find possible factors associated with high risk of mortality and chronic diseases are used as decision making tool. The way these tools work is by simulating the human brain functioning, with the greatest advantage in big data processing capabilities. This technology offers methods such as supervised learning based (Random Forest, Support Vector Machine, and Artificial Neural Network), unsupervised learning based (capable of finding patterns of unlabeled data and cluster), and hybrid methods based on trial and error (Reinforcement Learning) [9,10,11].

## 2. Materials and Methods

For the data analysis employed for this paper, we used Orange software version 3.23. This software offers a visual programming environment that allows analyzing data from statistics to machine learning by using interconnected “widgets” that indicate the flow that data must follow and functions applied to data. To analyze the database provided by Clinic Medical Norte to find the secondary variables to consider a thoracic pain with a cardiac origin, we used 17 widgets.

Five different machine learning algorithms available in the Orange data mining toolkit [12], including k-nearest neighbor (kNN), decision tree, support vector machine (SVM), random forest, and logistic regression, were employed in this study. To evaluate the classification models, we use a 10-fold cross-validation strategy, where the original samples were randomly partitioned into ten equal-sized subsamples, and we retained a single subsample as validation data for testing. For this analysis, we use the following tools of Orange:

Data
File: Allows to upload the file to analyze. When loading the file, the value of each variable must be selected, that is, whether it is categorical, numeric, or text, and its role within the analysis, if it works as a feature, target, meta, or skip. In this case, our aims were AMI and FRCV; we registered it as categorical.Data table: Allows us to visualize in a table the uploaded file.

Visualize
Scatter plot: This graphic allows us to see continuous data represented in two dimensions.Box plot: This graphic shows the distribution of the values of each attribute.Classification tree viewer: Allows us to visualize the resulting analysis of the model tree classification. It shows a classification tree that indicates the hierarchy of each value, which allows us to determine the most important.

Models
Classification Tree.Logistic regression.Random Forest.kNN.SVM.

Evaluation:
Tests and Scores: Analyzes the information using selected models, and shows different parameters like accuracy, Precision, F1, recall.Confusion Matrix: Generates a matrix presenting false positives, true positives, false negatives, and true negatives.

### 2.1. Description of the Database

The database (provided by the Clinic Medical Norte) contains 27 data items from 256 patients (See Appendix B). The average age of the participants included in this study is 60 years. Table 1 presents the assessment criteria used in the patients of the Clinic Medical Norte.

### 2.2. Machine Learning Models for Thoracic Pain Evaluation

Figure 1 shows the thoracic pain management guide. To create these models, we use the variables that provide post-disease information, such as medications. Furthermore, according to the clinical practice guideline, the variables used as a diagnosis were eliminated [14,15].

To identify the most influential variables in the different created models, a classification of these variables was done by assigning to each one a score, with the lower scores being indicative of greater importance. For this analysis, we considered a sample of 256 patients, and two machine learning techniques were used: Tree classification and cross-validation. For statistical analysis, the “distributions” tool from Orange was used.

## 3. Results

The database provided by Clinic Medical Norte is formed by 256 patients, of which 35.66% had an IAM. Of those who suffered from an IAM, 63.04% had dyslipidemia, 50% suffered from CKD, 71.74% had diabetes, 36.96% had Hypertension, 72.42% were smokers or smoke, and 54.35% were men.

### 3.1. Tree Classification

As mentioned before, in this model, the target was IAM where the decision tree suggested six factors to determine if the person with thoracic pain was in risk to present an IAM; these factors were found as the current considered in the emergency room when a patient with chest pain arrives. Another target examined was the variable of Risk Factors for Cardiovascular Disease (FRCV), this target was considered as a categorical variable, which showed if the patients suffered from a disease of had a negative result in clinical tests, and the result of the decision tree revealed the proposed secondary factors to evaluate if a thoracic pain has a cardiac origin or does not. In Table 2, the results from botch tree classification analysis are shown.

### 3.2. Cross-Validation

For the cross-validation analysis, 66% of the database was used to train the models, using a number of 10 folds as parameter. For this, the used classifiers were: tree classification, random forest, SVM, logistic regression and kNN. Considering the results of the FRCV decision tree, of the previous analysis, these secondary factors were used as targets, of which results are presented in Table 3.

In Table 4, the variables determined as secondary factors to consider when a patient arrives in the emergency room with chest pain are shown. The research suggests a close relation between these diseases and habits, since one can be caused by another. Among these variables, according to the results of the machine learning analysis, dyslipidemia may be considered as the main disease responsible for possible thoracic pain with cardiac origin, followed by hypertension, smoking habits, diabetes, chronic kidney disease, and PPT ranges. In the case of the variable dyslipidemia, the best obtained result was using logistic regression with an accuracy of 0.969, F1 of 0.938, precision of 0.937, and recall of 0.940. In hypertension, we found an accuracy of 0.994, F1 of 0.966, precision of 0.966, and recall of 0.966. For smoking, we found an accuracy of 0.918, F1 of 0.799, precision of 0.796, and recall of 0.803. Lastly, for diabetes, we found an accuracy of 0.986, F1 of 0.961, precision of 0.963, and recall of 0.961. For the variable of PPT ranges, the Random Forest model showed better results, with an accuracy of 0.977, F1 of 0.880, precision of 0.881, and recall of 0.891.

## 4. Discussion

In emergency rooms, between 5% and 15% of the patients report thoracic pain, whereby 23.8% of patients with thoracic pain are related to cardiovascular pathologies [16]. Another case that was found to be alarming in the Hospital de la Línea de la Concepción in Cádiz is that 25% of the patients present an AMI (Acute Myocardial Infarction) after they left the emergency room due to a normal electrocardiogram [4], which can be construed as 1 out of 4 patients had a wrong diagnosis, which could lead to a sudden death. For this reason, it is important to find new methods to efficiently classify the origin of thoracic pain, since it can be related to cardiogenic factors either ischemic or not ischemic; and not cardiogenic factors being of gastrointestinal, pulmonar, neuromuscular, or psychological origin [17]. Due to the multiple risk factors for CVD, it is critical to find the nearest linked factor to a sudden death caused by a cardiomyopathy with thoracic pain as a symptom, considering that health conditions and lifestyle, including alimentation, have a considerable impact in CVD development. For this, machine learning techniques and tools are proposed to predict cardiopathies that could lead to sudden death [4,18]. Furthermore, studies found that when a person presents various risk factors, the probabilities to develop a CVD in a 10-year range increases significatively [19]. Hence, it is recognized as widely important to identify the risk factors present when the patient arrives at the emergency room with thoracic pain. This study suggests the use of the risk factors obtained as results in the tree classification analysis and validated by cross-validation method, in the evaluation of the thoracic pain in order to classify it as cardiac or not cardiac, considering these as secondary factors alongside those currently used in the emergency rooms. However, these results must be interpreted with caution and a series of limitations must be taken into account since the study was carried out only with elderly patients and with less than 30 variables; to achieve more precise results in future studies, the use of a database with more variables to consider and a population with different age ranges is proposed, and with this, better training in machine learning models would be achieved, which would allow for finding greater differentiation between variables.

### 4.1. Relationship of Secondary Factors Variables with IAM

#### 4.1.1. Patients with Smoking Habits

Tobacco consumption increases the oxidative stress due to the free radicals generation for both passive and active consumers, and for this reason it is known as the main factor in the development of different diseases, including CVD. Among the adverse effects in health caused by tobacco consumption besides oxidative stress, studies found a relationship in the increase of the arterial pressure and cardiac frequency, increase in inflammation, developments of atherosclerosis, thrombosis, and damage in both arterial coronary systems [20,21].

Regarding chest pain, a study done with 70,208 participants, which mostly have smoking habits, discusses an experimentation using methods as pain tolerance testing and surveys, which concluded that people with smoking habits tend to have lower pain tolerance; this information is important to know regardless that the intensity has not relation with a cardiac origin pain. Moreover, it was found that the chest pain in smoking patients can be originated by, inter alia, the frequency in tobacco consumption, chronic cough and shortness of breath [22].

#### 4.1.2. Patients with Hypertension

Hypertension is one of the most important risk factors on CVD. Worldwide, hypertension is responsible for 54% of strokes, and 47% of ischemic cardiopathy [23]. It has also been observed that after a decade of presenting hypertension, the risk of contracting any CVD has increased from 15% to 30% [24]. On the other hand, evidence of a study made in 1997 in Chile found interesting results in records of people with obesity, which suggest that obesity increases blood pressure with 6.5 mmHg, plasma cholesterol with 12 mg/dL, and 2 mg/dL of blood glucose for each 10% of accession in the patient’s weight [25].

A study described in the Cuban Magazine of Health compares their findings done between 2007 and 2011 with findings made in Spain on 2011, and both results agree with the fact that Hypertension is strongly related with sudden death by a cardiac event; this, due to the development of an adaptive process initiated by blood pressure causing hypertrophy as a result of left ventricular injure. It is also stated that a combination of hypertension with smoking habits or any other risk factor as diabetes, dyslipidemia, and obesity can lead to an increase in the left ventricular hypertrophy expanding the probabilities of suffering a cardiac event [26].

#### 4.1.3. Patients with Diabetes

Diabetes is a disease that is also tightly related with CVD and obesity, when there are no other risk factors involved it is called Diabetic Heart Disease (DHD). Amidst the possible factors of the relationships between these conditions, insulin resistance, hyperglycemia, and hyperinsulinemia were found to be responsible for the decrease in elasticity of the tissue generated by an impact in the production of collagen, which provokes myocardial damage leading to hypertrophy and fibrosis [27].

Despite the fact that in a study carried out in the Grama region, it was determined that patients with DM II who present other cardiovascular risk factors, compared to those without Diabetes, did not present chest pain as a symptom. However, the study suggests that those with DM II are exposed to cardiac failure by a factor of 2.8, since it has also been found that patients with this disease suffer from alterations in diastolic function without having any history of cardiovascular disease [28].

The chemical reactions generated by cardiac metabolism are oxidative in nature, so that as there is a lack of biological contribution to the region of cardiac tissue, ATP is stopped in cardiomyocytes, which in turn, causes a metabolic change due to the lack of oxygen and nutrients directly affecting cardiac functionality [29].

#### 4.1.4. Patients with Chronic Kidney Disease

Chronic Kidney Disease (CRD) is another risk factor linked to CVD. Findings from a study made with dialysis patients revealed that CVD patients start their development in precocious phases of the CRD, causing problems such as left ventricular hypertrophy, atherosclerosis, and vascular calcifications [30]; therefore, early detection and treatment of this disease can reduce the chances of death from CVD, as well as decrease kidney damage, since it was revealed in a study carried out using patients with advanced ECR with and without dialysis, which those with an AMI have a very low chance of survival [31,32]. On the other hand, CKD is found in some cases related to diabetes, which is called diabetic nephropathy, which develops hypertension and kidney damage [33].

#### 4.1.5. Patients with Dyslipidemia

Among the distributions related to patients with dyslipidemia and pain in the database from Medical Norte, 22.87% of the patients with dyslipidemia presented soft pain, while 26.74% presented moderate pain and 14.34% presented severe pain. Of the remaining individuals without dyslipidemia, only 5.81% presented severe pain. Despite these results, it is important to know, beyond pain, how dyslipidemia would affect the cardiovascular system.

Dyslipidemia is a disease where the regulation of lipids in blood is affected by the augmentation of cholesterol and triglycerides, which in turn produces the accumulation of lipids in the arterial walls causing ischemic heart disease, which can lead to death; the main reason of this disease is due to obesity, even though it can be also a genetic disease [34]. The most known disease in Mexico is obesity since, in 2012, 71.3% of the population was diagnosed with obesity, while in Baja California, 74.9% of the population presented obesity and overweight [35]. Obesity is one of the main factors for various diseases, including CVD. The relationship between dyslipidemia and obesity is very close due to the excess of fatty tissue, which produces an insulin resistance [36]; also, it is related to diseases such as Diabetes Mellitus II (DM II). According to the WHO in 2012, 44% of the people living in Baja California developed DM II due to obesity and overweight, pathology related with hypertension, dyslipidemia, CDV, osteoarthritis, and different types of cancer [37].

This documental research confirms the correlation between the proposed secondary risk factors related with possible thoracic pain with cardiac origin. In Figure 1, the diagram above shows graphically the relation between these variables, which was confirmed by both the assessment with machine learning and bibliography. The figure is divided into three main components, the blue navy hexagon in the center indicates the target, which is thoracic pain with cardiac origin, the second level with blue hexagons shows the six main conditions proposed as factors to consider in the determination of a cardiac event with thoracic pain as symptom, and the last level with light blue hexagons shows some effects that the main factors have in health. The orange lines used in Figure 2 express the relationship between the conditions.

## 5. Conclusions

Among the main health problems presented in the country are deaths from obesity problems and cardiovascular diseases, which in turn are related to each other, sharing other risk factors. When considering cardiovascular problems as diseases that can cause sudden events involving a person’s life, it is important to learn to recognize the patterns that these cardiac events present and to take into account the factors that have the greatest impact on their development. It is known that in emergency rooms, there are a limited number of patients to attend, and since thoracic pain is a symptom of a future cardiac event, but also a symptom of different diseases, it is important to learn to recognize when thoracic pain is of cardiac origin and non-cardiac.

Nowadays, there are different computer tools such as machine learning, deep learning, and artificial intelligence, which, through algorithms, can find patterns and classify a large number of data. This is why it was decided to carry out a machine learning analysis of a database provided by Clinic Medical Norte in Tijuana, Baja California, Mexico. The results of this analysis suggest variables that can be considered secondary conditions to classify thoracic pain as cardiac in addition to those already established in the emergency department, such as Troponin levels, smoking habits, and diseases such as dyslipidemia, chronic kidney disease, diabetes, and hypertension.

## Figures and Tables

**Figure 1 ijerph-18-02155-f001:**
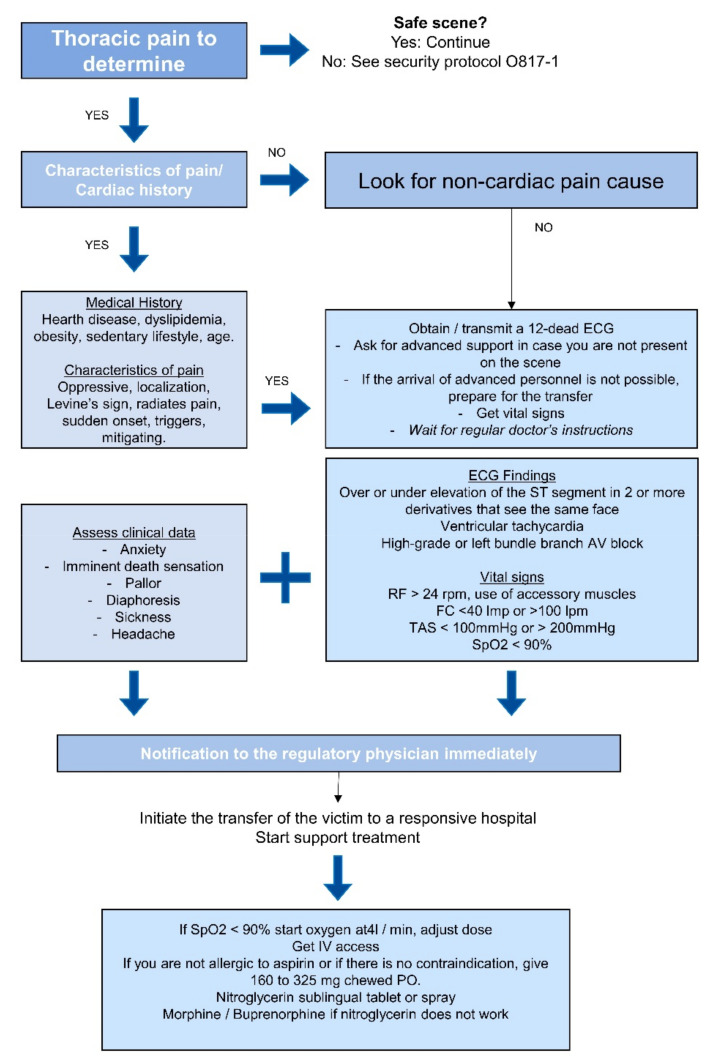
Thoracic pain management guide.

**Figure 2 ijerph-18-02155-f002:**
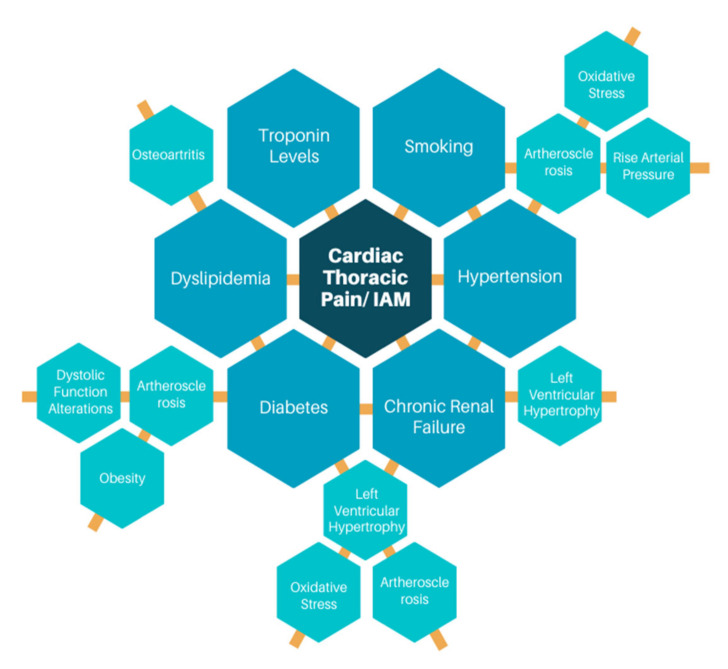
Relationship between secondary risk factors variables.

**Table 1 ijerph-18-02155-t001:** Assessment criteria used for patients.

Classic Patterns of Thoracic Pain				
Condition	Location	Radiating Pain	Duration	Type of Pain
AMI	Retroesternal	Arm, Neck	>15 min	Oppressive
Angina	Retroesternal	Arm, Neck	5–20 min	Oppressive
Aortic dissection	Retroesternal	Interescapular	Constant	Tearing
TEP	Hemithorax	-	Constant	
Pneumothorax	Hemithorax	Neck, Back	Constant	
Pericarditis	Retrosternal, shoulder, arm	Back, Neck	Constant	
Esophageal ruptura	Retrosternal	Posterior Thorax	Constant	
Esofagitis	Retrosternal	Interescapular	Minutes to hours	
Esophageal spasm	Retrosternal	Interescapular	Minutes to hours	
Musculoskeletal	Localized	-	Variable	

Note. Adapted from *Prehospital Medical Emergency Manual* (p. 334), by A. Pacheco-Rodríguez, A. Serrano-Moraza, J. Ortega-Carnicer, F. Hermoso-Gadeo, 2001 [13], Madrid, España: Aran Ediciones. Copyright 2001 by Aran Editions.

**Table 2 ijerph-18-02155-t002:** Comparison between Acute Myocardial Infarction (AMI) and Risk Factors for Cardiovascular Disease (FRCV) used as a target for Tree classification.

Tree Classification		
Level	AMI	FRCV
1	Angio TAC	Dyslipidemia
2	Ergometry	CKD
2	TnT curve 4 h	Diabetes
3	Eco-Stress	Hypertension
3	Catheterization	Smoking habits
4	PPT	Age
5	-	TnT entry
6	-	Gender

**Table 3 ijerph-18-02155-t003:** Metric’s formulas.

Metric	Expresion
Accuracy	$Accuracy=\frac{True\;positive+True\;Negative}{True\;Positive+True\;Negative+False\;Positive+False\;Negative}$
F1	$F=2*\frac{Precision*Recall}{Precision+Recall}$
Precision	$Precision=\frac{True\;positive}{True\;Positive+False\;Positive}$
Recall	$Recall=\frac{True\;Positive}{True\;positive+False\;Negative}$

**Table 4 ijerph-18-02155-t004:** Cross-Validation results using a target FRCV decision tree results.

Variables	Classification	Accuracy	F1	Precision	Recall
Dyslipidemia	Tree classification	0.780	0.787	0.787	0.787
	SVM	0.823	0.737	0.750	0.753
	kNN	0.630	0.618	0.614	0.622
	Logistic Regression	0.969	0.938	0.937	0.940
	Random Forest	0.795	0.753	0.614	0.622
Hypertension	Tree classification	0.765	0.762	0.761	0.762
	SVM	0.846	0.757	0.757	0.758
	kNN	0.733	0.689	0.688	0.691
	Logistic Regression	0.994	0.966	0.966	0.966
	Random Forest	0.825	0.762	0.762	0.764
Smoking	Tree classification	0.691	0.580	0.578	0.586
	SVM	0.716	0.514	0.547	0.569
	kNN	0.658	0.510	0.504	0.532
	Logistic Regression	0.918	0.799	0.796	0.803
	Random Forest	0.739	0.587	0.585	0.606
Diabetes	Tree classification	0.712	0.727	0.729	0.701
	SVM	0.746	0.612	0.725	0.705
	kNN	0.546	0.602	0.590	0.625
	Logistic Regression	0.986	0.961	0.963	0.961
	Random Forest	0.733	0.704	0.706	0.724
Rangos PPT	Tree classification	0.997	0.990	0.990	0.990
	SVM	0.895	0.707	0.699	0.720
	kNN	0.992	0.855	0.954	0.960
	Logistic Regression	0.951	0.845	0.841	0.851
	Random Forest	0.977	0.880	0.881	0.891

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
