# Peer review of "Assessment of Thoracic Pain Using Machine Learning: A Case Study from Baja California, Mexico"

_ijerph, 2021, doi:10.3390/ijerph18042155_

Round 1

Reviewer 1 Report

All issues have been well addressed. There is no major comment to raise. However, this reviewer could not find the answer to comment 4 (in the authors' response file). 

Author Response

Dear Editor,

Thank you for the opportunity to revise our manuscript, Assessment of thoracic pain using machine learning: a case study from Baja California, Mexico. We appreciate the careful review and constructive suggestions. Please find attached the answer to your comments.

Thank you for your consideration.

Sincerely,

Reviewer 2 Report

This is an extensive analysis of associations between specific variables and thoracic pain of cardiac origin with the use of of machine learning techniques. In general the study is well composed, however some sentences are too complex or incomplete. For instance see Introduction: lines 76-82 and Discussion: lines 210-220. 

Author Response

Dear Editor,

Thank you for the opportunity to revise our manuscript, Assessment of thoracic pain using machine learning: a case study from Baja California, Mexico. We appreciate the careful review and constructive suggestions. Please, find attached the corrections. 

Best regards, 

Reviewer 3 Report

In this manuscript entitled "Assessment of thoracic pain using machine learning: a case study from Baja California, Mexico" the authors used machine learning techniques, performing an analysis of 27 variables provided by a database with information from 256 patients from a private Hospital in Tijuana, Mexico. 

The manuscript is very interesting. However, some issues should be addressed by the authors:

ABSTRACT

  • Aim is not clear presented
  • Incude more statistical results

INTRODUCTION

  • Aim is not clear presented. Also with the study hypothesis.
  • Improve the rationally from the study

METHODS

  • Please, include a clear statement about the study design.
  • Start the Method section with the Description of the database and the pacients. It is better than start direct with the machne learning analysis.
  • Stattistical analysis is not understandable and clearly described. Please, improve this section.

DISCUSSION

  • Figure 2 is not presented in the text. The authors always have writting abou the Figure 1. Please, present and describe the Figure 2.
  • Limitation should be presented by the authors

Author Response

Dear Editor,

  Thank you for the opportunity to revise our manuscript, Assessment of thoracic pain using machine learning: a case study from Baja California, Mexico. We appreciate the careful review and constructive suggestions. Please, find attached the answer to your revision. 

Round 2

Reviewer 3 Report

All my comments were addressed.

Congrats.

This manuscript is a resubmission of an earlier submission. The following is a list of the peer review reports and author responses from that submission.

Round 1

Reviewer 1 Report

The authors provided the machine learning technique to determine the correlated variables with thoracic pain of cardiac origin. The study interesting but has very low readability, so it is strongly suggested that the following issues should be addressed.

1. Introduction: It is unclear what the ultimate goal of this study is. It should be provided what problems are currently being raised about present methods to assess in patients with CVD (including emergency), and the benefits of clinical contribution through the method the authors' arguing (maybe software orange). Further, it should also be provided the purpose and reason for its use. It is very ambiguous whether the content of lines 51-73 should be included.
2. Methods: It is wondered if the statistical analysis method did not need to be presented.
3. Results: Lists parts that cannot be inferred at all from the contents. The contents that resolve the following questions must be described somewhere in the paper.
- What is the relevance of Figure 1 to this study?
- What is the F1 group?
- How to calculate “Accuracy” and other values?
- What does the mean of "Accuracy" values in the classification?
- So, how to find the basis for the determining correlation variable with thoracic pain of cardiac origin using machine learning technology?
4. Discussion: If the secondary risk factors were gathered by an effort to analyze with the tables, this statement is close to the results. In the discussion section, it would be better to summarize the entire study and describe its limitations and clinically applicable benefits through this study, importantly include what is the key technique to determine each variable to recognize cardiac or non-cardiac origin in thoracic pain.

Reviewer 2 Report

Authors analize 27 variables from 256 patients using tree classification and cross validation in order to identify when thoracic pain is of cardiac origin.

The introduction contains many old references (Ref 1 for example). The introduction lacks a concrete and logical order, which makes it very difficult to read. It also contains phrases that the way they are written also make it very difficult to understand its meaning (lines 48-50), or some basical concepts are not correctly explained (line 53). 

The work uses a "data base provided by a private hospital", but does not mention the name of the hospital, does not indicate how, when or for how long the patients were recluited. 

There is not a basal description of the cohort. 

In my opinion, many basic data are missing to be considered for review or publication.